# Inflow Prediction of Centralized Reservoir for the Operation of Pump Station in Urban Drainage Systems Using Improved Multilayer Perceptron Using Existing Optimizers Combined with Metaheuristic Optimization Algorithms

Eui Hoon Lee

School of Civil Engineering, Chungbuk National University, Cheongju 28644, Republic of Korea; hydrohydro@chungbuk.ac.kr

**Abstract:** Owing to the recent increase in abnormal climate, various structural measures including structural and non-structural approaches have been proposed for the prevention of potential water disasters. As a non-structural measure, fast and safe drainage is an essential preemptive operation of a drainage facility, including a centralized reservoir (CRs). To achieve such a preemptive operation, it is necessary to predict the inflow of the drainage facilities. Among the drainage facilities, CRs are located downstream of the drainage area, and their pump stations are operated according to the CR water level. The water level of a CR depends on the inflow, as does the preemptive operation of its pump station. In this study, as a nonstructural measure, the inflow prediction for the CR operation in an urban drainage system was proposed. For predicting the inflow of a CR, a new multilayer perceptron (MLP) using existing optimizers combined with a self-adaptive metaheuristic optimization algorithm, such as an improved harmony search, was proposed. Compared with the adaptive moment, which yields the best results among other existing optimizers, an MLP using an existing optimizer combined with an improved harmony search improves the mean square error and mean absolute error by 0.1767 and 0.0349, respectively.

**Keywords:** centralized reservoir; urban drainage system; multilayer perceptron; metaheuristic optimization algorithm; improved harmony search

## 1. Introduction

To prevent disaster caused by water, structural measures focusing on increasing the capacity of hydraulic facilities, such as pump stations, centralized reservoirs (CRs), decentralized reservoirs (DRs), and pipes, have been proposed. However, owing to the rapidly increasing frequency of localized heavy rainfall, implementable structural measures are limited by time and cost. Hence, non-structural measures, such as the operation of urban drainage systems (UDSs), should be implemented. Among the various non-structural measures available, the operation of drainage facilities in a UDS is important for reducing flood damage. A preemptive operation is essential for the fast and safe drainage of drainage facilities, such as a CR. The operation of the pump stations at a CR depends on the CR water level, and to achieve a preemptive operation, the inflow directly connected to the water level should be predicted. Additionally, an accurate inflow prediction of a CR is essential for the operation of the pump stations. A multilayer perceptron (MLP) with existing optimizers and combined with metaheuristic optimization algorithms has been suggested to predict the inflow of a CR.

A perceptron, which is a type of artificial neural network (ANN), was developed based on the concept of a hypothetical nervous system and the memory storage of the human brain [1]. The initial perceptron was a single-layer version with the ability to solve only problems that allow linear separations. Hence, an MLP was developed to overcome the

abovementioned limitation [2]. Henceforth, various ANN-related approaches, including those pertaining to MLPs, have been proposed. A recurrent neural network (RNN) using error backpropagation was developed for the construction of appropriate internal representations [3]. Additionally, a convolutional neural network (CNN) designed to process two-dimensional shapes was developed for document recognition [4]. Long short-term memory (LSTM), which is a type of RNN, was developed to solve the vanishing gradient problem of a conventional RNN [5]. To solve the shortcomings of LSTM, which requires a significant amount of memory, a gated recurrent unit (GRU) with a fast computational structure simplifying the model structure was proposed [6].

ANNs have been extensively used in investigations pertaining to hydrology and water resources. A multi-output neural network has been recommended for flow duration curve prediction and compared with single-output neural networks [7]. Approaches to the utilization of applicable deep-learning methods for future water resources have also been recommended [8]. Flood susceptibility mapping is a hybrid model that combines a swarm intelligence algorithm with a deep neural network [9]. A standardized streamflow index for hydrological drought using support vector regression, gene expression programming, and M5 model trees has also been suggested [10]. Considering a state-of-the art modeling framework appraisal, artificial intelligence models have been used to predict suspended river sediment transport for future research directions [11]. The spatial pattern of saturated hydraulic conductivity was predicted using a novel genetic algorithm (GA) based hybrid machine learning pedotransfer function [12]. Metaheuristic optimization algorithms, such as the swarm intelligence algorithm, have also been used to improve the performance of an ANN.

Heuristics are simple inference methods that can be promptly used in situations where rational decisions cannot be realized owing to insufficient time or information. It is difficult to develop heuristics when only the characteristics of each problem to be solved are available. Metaheuristics are high-level heuristics that are applicable to various problems without being restricted by the information regarding a specific problem. Although various metaheuristic optimization algorithms exhibit different characteristics, their concept and theory are simple and offer an excellent solution search ability. They can therefore be applied to engineering, sciences, business, and social sciences.

The first metaheuristic optimization algorithm, that is, the genetic algorithm (GA), was proposed [13]. The GA was created by mimicking the crossover and mutation processes in evolutionary theory, after which various notable metaheuristic optimization algorithms were developed, as described in the following. Ant colony optimization is based on the behavior of ants [14]. As an ant identifies a path leading to a food source from its home, it releases pheromones, which rapidly evaporate. However, if the path is optimal, most ants will select it and the pheromone will remain. Particle swarm optimization (PSO) was inspired by the movement of individuals residing in groups, such as ants, fish, and birds. PSO is an algorithm that gradually identifies an optimal solution by comprehensively considering the current velocity of particles and the velocity at which the states of other particles are considered. Harmony search (HS) was developed to encourage improvisation among musicians [15]. The purpose of HS is to identify the best harmony by combining notes produced by each musician. HS has been used extensively owing to its simple structure. Although demonstrating a good performance, the local search for numerical problems is limited when applying such a search. Hence, an improved version called improved harmony search (IHS) has been proposed. IHS, which improves the fine-tuning function based on HS, has demonstrated an excellent performance in various optimization problems [16].

Moreover, hydrological predictions, such as water level and runoff predictions, have been investigated by combining metaheuristic optimization algorithms with MLPs. In fact, researchers have proposed applying the GA as a metaheuristic optimization algorithm in combination with an MLP. Rainfall-runoff modeling was conducted by combining a real-coded genetic algorithm (RCGA) with an MLP [17]. An MLP coupled with an RCGA

was suggested for predicting cumulative and discrete rainfall in the Upper Parramatta River basin of Sydney, Australia [18]. An MLP combined with an RCGA was applied to forecast daily rainfall-runoff in the Ourika basin, Morocco [19]. An MLP–RGA hybrid model demonstrated the highest accuracy in water level prediction for the Nakdong River Basin, Korea [20]. In a recent study, HS was applied as a metaheuristic optimization algorithm in combination with an MLP. An MLP combined with HS has also been presented to predict the water level of an urban stream based on pump station discharge [21]. In the abovementioned studies, only the water level or runoff of streams and rivers was predicted. No study has been conducted on the prediction of inflow in urban drainage system. Additionally, minutely prediction for the operation of urban drainage systems were not conducted.

Among non-structural measures, the inflow prediction of a CR for the pump station operation in a UDS is suggested in this study. The inflow prediction of a CR enables the preemptive operation of a pump station. The pump station can be quickly drained based on the inflow prediction of the CR, and it is possible to secure additional storage capacity by maintaining a low CR water level. A new optimizer was proposed by combining IHS, an improved HS that has shown a good performance among various metaheuristic optimization algorithms, with existing optimizers. MLP combined with the new optimizer was applied to the inflow prediction of a CR, the results of which were compared with those of existing optimizers alone and existing optimizers combined with HS.

The characteristics of this study can be divided into four main categories. The first is the selection of monitoring nodes used to consider the state of stormwater conduits. The second is the construction of training data used for an MLP, an MLP combined with HS (MLPHS), and an MLP combined with IHS (MLPIHS). The third is the preprocessing of the training data. The fourth is the inflow prediction of a CR and a comparison of results from each MLP.

## 2. Methodologies

### 2.1. Overview

Rainfall, water level data on the first/maximum flooding nodes, and the CR inflow from 2010 to 2020 were acquired. From 2010 to 2019, learning data for rainfall, the water level of the first/maximum flooding nodes, and the CR inflow were collected for the inflow prediction of CR in 2020. After creating the training data, training was conducted by building and applying an MLP, MLPHS, and MLPIHS. The CR inflow was predicted using the rainfall and water level data of the first/maximum flooding nodes as input data. In this study, the applied MLP was constructed using TensorFlow [22]. Figure 1 shows the workflow of this study.

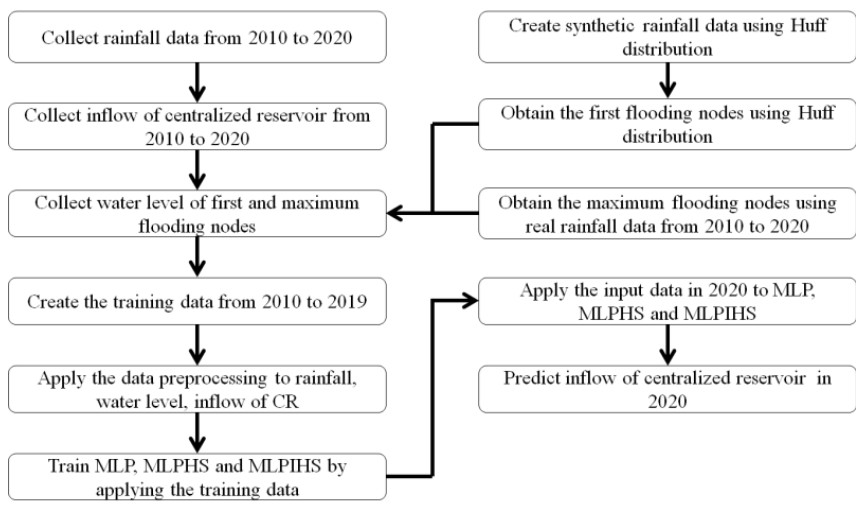

**Figure 1.** Workflow of this study.

### 2.2. Preparation of Training Data

The training data must be pre-processed for an accurate prediction of the CR inflow. Although various data preprocessing methods for learning purposes are available, a correlation analysis and normalization were applied in this study.

#### 2.2.1. Correlation Analysis

A correlation analysis was applied to consider the time difference between the input data (rainfall and water level of the first/maximum flooding nodes) and the target data (CR inflow). Equation (1) was used to calculate the correlation coefficient:

$$r_{x,y} = \frac{\sum_{i=1}^{n}(x_i - \overline{x})(y_i - \overline{y})}{\sqrt{\sum_{i=1}^{n}(x_i - \overline{x})^2}\sqrt{\sum_{i=1}^{n}(y_i - \overline{y})^2}} \tag{1}$$

where $r_{x,y}$ is the correlation coefficient, $x_i$ is the input data, $y_i$ is the target data, $\overline{x}$ is the average of $x_i$, and $\overline{y}$ is the average of $y_i$. Figure 2 shows the application of a correlation analysis considering the lag time.

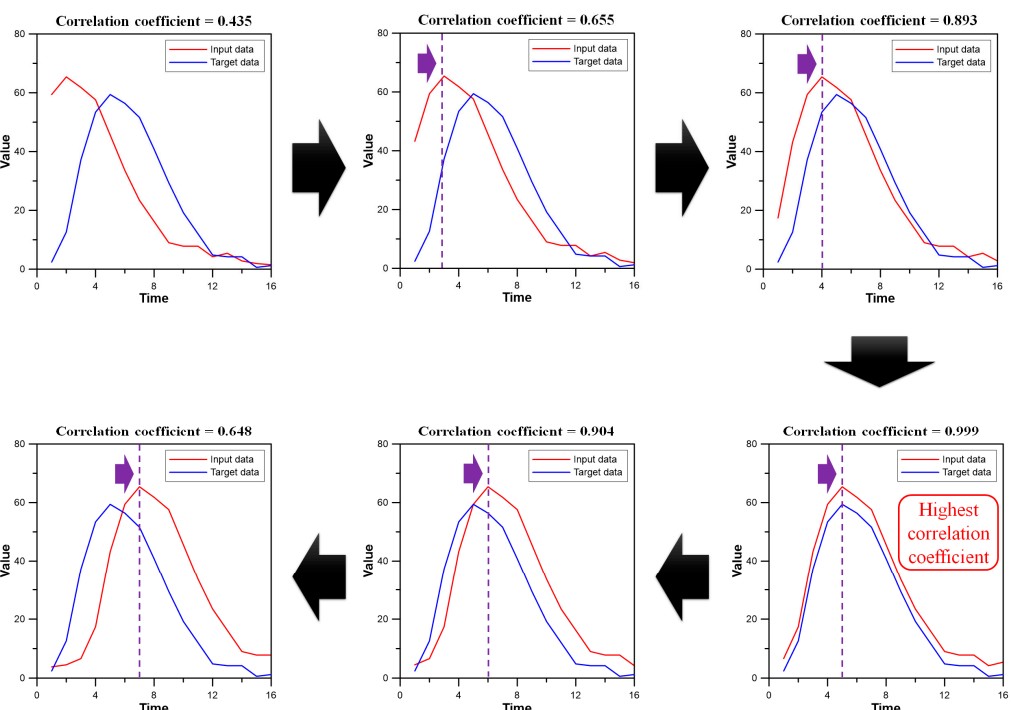

**Figure 2.** Application of a correlation analysis considering the lag time.

#### 2.2.2. Normalization of Training Data

A study on normalization was conducted when applying ANN to rainfall-runoff models [23]. When the MLP is trained using data with a wide range of values, the prediction performance can degrade owing to the difference between the input and target data. Hence, the data should be converted into values between 0 and 1 through normalization. Normalization was conducted for all data for each input data and target data. The maximum value of each data was converted to 1 and the minimum value of each data was converted to 0. Equation (2) expresses the normalization:

$$y_i = \frac{(x_i - x_a)}{(x_b - x_a)} \tag{2}$$

where $y_i$ is the normalized value, $x_i$ is the observed value, $x_b$ and $x_a$ are the maximum and minimum observed values, respectively.

*2.3. MLP Combined with IHS*

The MLP is the most basic type of an ANN and comprises one input layer, one or more hidden layers, and one output layer. The weight and bias are set as parameters, and they can be used to express non-linear problems. Figure 3 shows the structure of the MLP including MLPHS and MLPIHS used in this study.

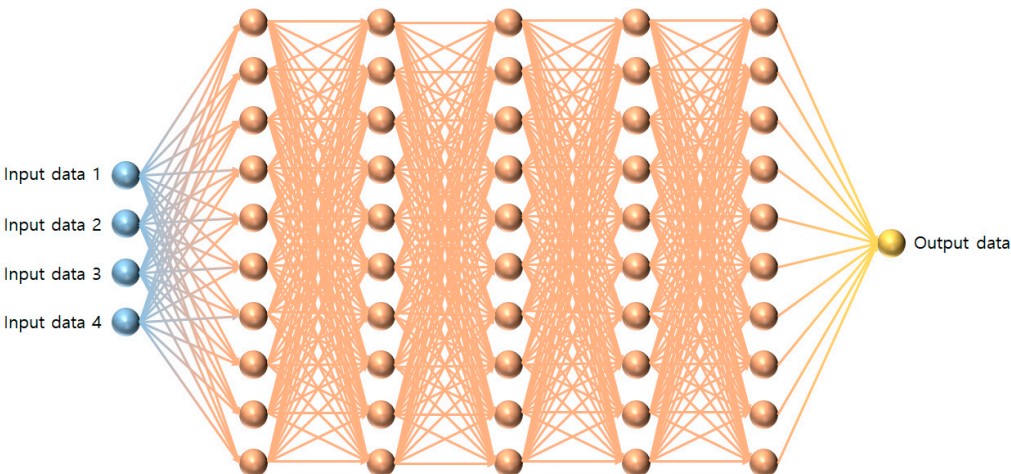

**Figure 3.** Structure of the MLP including MLPHS and MLPIHS used in this study.

The input and output layers shown in Figure 3, comprise four input data and one output data, respectively. Five hidden layers are used, and each hidden layer contains ten nodes. For an MLP using the existing optimizers, a rectified linear unit (Relu) was applied as the activation function. Including MLPHS and MLPIHS, there were 10,000 epochs for all MLPs.

2.3.1. Existing Optimizers in MLP

The optimizer applied in the MLP was used to compute the weights and biases between nodes. The most basic optimizer is gradient descent (GD), in which a differentiation is conducted to obtain the weights and biases between nodes; however, this entails the use of all available data. Hence, stochastic gradient descent (SGD) was introduced to overcome the shortcomings of GD. Additionally, momentum when considering the inertia was proposed to improve GD, and the Nesterov accelerated gradient (NAG) was suggested to improve the momentum. With SGD, data are randomly selected, whereas with GD, the learning performance differs depending on the learning rate. An adaptive gradient (Adagrad) was proposed to overcome the shortcomings of SGD. With Adagrad, the learning rate can reach zero during the learning process through the application a flexible learning rate based on the learning process, which prevents learning from taking place. To overcome the shortcomings of Adagrad, root mean squared propagation (RMSprop) is an optimizer that improves the learning stoppage using an exponential average, whereas adaptive delta (Adadelta) updates the learning rate using a Hessian matrix and an exponential average. Moreover, the adaptive moment (Adam), a combination of momentum and RMSprop, updates the learning rate using the exponential mean and square of the weights. Adamax, an extension of Adam applying a new infinity norm, has been proposed. Follow the regularized leader (Ftrl), an algorithm that includes normalization of follow the leader by considering the gradient (leader) with the smallest loss, has been proposed. Nadam, which is a combination of NAG and RMSprop, identifies new weights and biases at new locations after propagating in the momentum direction. The existing optimizers applied to the MLP in this study were GD, SGD, Adagrad, RMSprop, Adadelta, Adam, and Nadam. The existing optimizers applied to the MLP in this study were Adadelta, Adagrad, Adam, Adamax, Ftrl, Nadam, RMSprop and SGD. Table 1 shows the existing optimizers in this study.

**Table 1.** Existing optimizers in this study.

| Optimizers | Description |
|---|---|
| Adadelta | Update the learning rate using a Hessian matrix and exponential average |
| Adagrad | Use a flexible learning rate based on the learning process |
| Adam | Combine momentum and RMSporp |
| Adamax | Apply a new infinity norm |
| Ftrl | Normalize the follow the leader by considering gradient (leader) with the smallest loss |
| Nadam | Combine Nesterov accelerated gradient and RMSprop |
| RMSprop | Improve learning stopping using an exponential average |
| SGD | Select randomly from the entire data set |

### 2.3.2. IHS

HS is a metaheuristic optimization algorithm proposed to encourage improvisation among musicians. The parameters used in HS are the harmony memory size (HMS), harmony memory considering rate (HMCR), pitch adjusting rate (PAR), and bandwidth (BW). HMS refers to the size of the harmony memory (HM) and is the number of candidate solutions that can be stored. HMCR is the probability of randomly selecting decision variables in HM to create a new combination of such variables. *PAR* refers to the probability of adjusting the decision variable selected through the HMCR when using the *BW*.

IHS is a metaheuristic optimization algorithm with an improved performance achieved by changing the *PAR* and *BW* for a local search in HS based on the number of iterations. In the latter iterations of the metaheuristic optimization algorithm, the effect of a local search, in which a detailed search can be conducted instead of a global search, becomes more important. Equation (3) shows the *PAR* of IHS:

$$PAR = PAR_{min} + (PAR_{max} - PAR_{min}) \times \left( \frac{I_c}{I_t} \right) \tag{3}$$

where $PAR_{min}$ and $PAR_{max}$ are the lower and upper boundaries of *PAR*, respectively; $I_c$ is the number of current iterations; and $I_t$ is the total number of iterations. *BW*, which is the range of a local search in HS, affects the results. Equation (4) shows the expression of *BW* in IHS:

$$BW = BW_{max} + exp\left( \ln\left( \frac{BW_{min}}{BW_{max}} \right) \times \left( \frac{I_c}{I_t} \right) \right) \tag{4}$$

where $BW_{min}$ and $BW_{max}$ are the lower and upper boundaries of *BW*, respectively. The IHS procedure is as follows:

Step 1. Create initial solutions based on the range of decision variables and generate the HMS.
Step 2. Sort the HM in HMS based on the value of the objective function.
Step 3(a). Select decision variables in the existing HM when HMCR is applied.
Step 3(b). Adjust new decision variables based on *BW* when *PAR* is applied.
Step 4. Create a new solution using the decision variables created in Steps 3(a) and 3(b).
Step 5. Compare the new solution with the worst solution in the existing HM to decide whether it should be replaced.
Step 6. Repeat Steps 2 to 5 until the termination criteria is satisfied.

In this study, a sensitivity analysis was conducted to establish the HS parameters. For MLPIHS, a value of 50 was applied for HMS; 0.85, for HMCR; 0.9, for $PAR_{max}$; 0.1, for $PAR_{min}$; 0.01, for $BW_{max}$; and 0.001, for $BW_{min}$.

### 2.3.3. Combined Optimizer Using Metaheuristic Optimization in MLP

An MLP using HS was recently proposed and applied to predict the runoff in urban streams [21]. Using the MLP proposed herein, a new MLP (MLPIHS) using optimizers combined with IHS has been suggested. Figure 4 shows a calculation flowchart for the MLP using optimizers combined with HS and IHS.

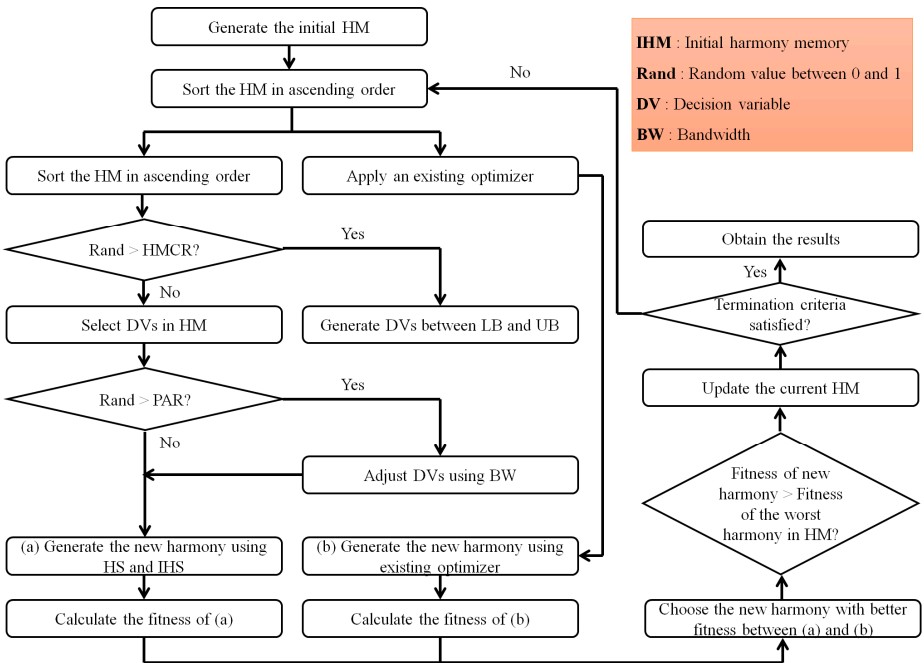

**Figure 4.** Calculation flowchart for MLP using optimizers combined with HS and IHS.

### 2.4. Selection of Monitoring Nodes

#### 2.4.1. Selection of Maximum Flooding Nodes

The maximum flooding nodes were selected based on the historical rainfall events within the study area. This method is used to find the node with the maximum flooding by applying the rainfall events from 2010 to 2019 as training data. The rainfall-runoff model for each rainfall event was required to obtain the maximum flooding nodes. A rainfall-runoff simulation was conducted using the storm water management model (SWMM) [24]. The selection of the maximum flooding nodes was used in a study on ways to operate drainage facilities for finding the most likely flooding node within the study area [25].

#### 2.4.2. Selection of First Flooding Nodes

The first flooding nodes were selected based on the Huff distribution [26]. The Huff distribution was segmented into four quartiles based on the location of the peak rainfall. The peak value of rainfall in the Huff distribution was located at the 0–25% duration in the first quartile, 25–50% duration in the second quartile, 50–75% duration in the third quartile, and 75–100% duration in the fourth quartile. The third quartile of the Huff distribution, proposed as the appropriate rainfall distribution in Korea, was selected [27]. By applying the third quartile of the Huff distribution, the total amount of rainfall increased by 1 mm, and this process was repeated until the first surcharge/flooding occurred. The simulation was conducted using the storm water management model (SWMM) [24]. Generally, the first flooding occurred between branch conduits; however, because it was difficult to represent the entire drainage system, the first flooding nodes between the main conduits were selected. The branch and main conduits were classified based on the sub-catchment area. If the area of sub-catchment exceeds 0.12 km$^2$, it is classified as a main conduit; otherwise, it is classified as a branch conduit [25]. For the applied duration, the concentration time within the study area was applied up to three times.

## 3. Application and Results

### 3.1. Study Area

The Han River passes through the center of Seoul, the capital city of the Republic of Korea, which is used as the study area in this study. The Han River comprises various tributaries, including the Anyang stream. Additionally, the Anyang stream comprises various tributaries, including the Dorim stream. Figure 5 shows the location of the study area and the SWMM network.

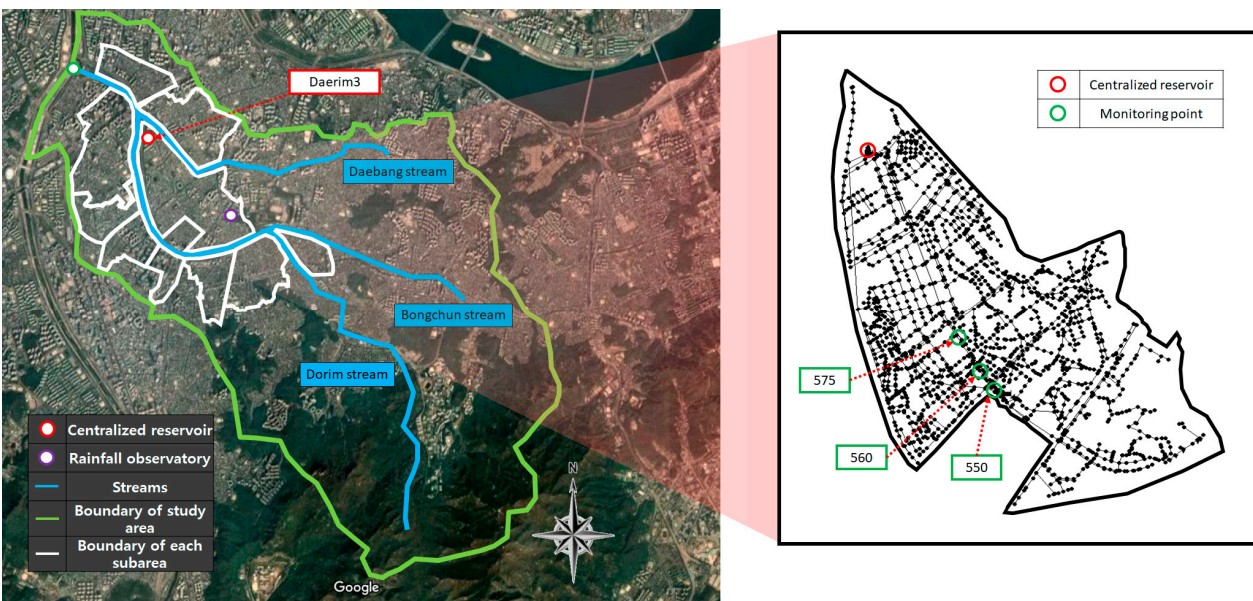

**Figure 5.** Location of study area and network of SWMM.

Among the 25 districts of Seoul, the Dorim stream flows adjacent to Gwanak-gu, Dongjak-gu, Guro-gu, and Yeongdeungpo-gu. The Dorim stream comprises two tributaries (Daebang and Bongcheon streams) with a length of 11 km and a watershed area of 41.93 km$^2$. Eleven pump stations are installed at the Dorim Stream, and the Daerim3 pump station has the largest drainage area. The Daerim3 pump station comprises 12 pumps (with a drainage capacity of 57.02 m$^3$/s) and has a drainage area of 2.49 km$^2$. The Daerim3 pump station has a CR that can receive inflow from the drainage area, the capacity of which is 36,200 m$^3$. In the network shown on the right side of Figure 5, the CR is located downstream of the drainage area. Additionally, the first and maximum flooding nodes selected in this study are indicated.

### 3.2. Preparation of Data for Inflow Prediction of CR

Training data must be prepared to predict the inflow of CR. The training data comprise the amount of rainfall, water level of the monitoring nodes, and CR inflow. Data on historical rainfall that occurred within the study area over the past decade were selected. Historical rainfall in this study means that caused flood damage in the study area. Data from 2015 and 2017 were excluded because historical rainfall occurred within the study area. Rainfall records from 2010 to 2020 were obtained from the Korea Meteorological Administration. The CR inflow was obtained based on the water level record of the CR and pump operation records at the Daerim3 pump station. Figure 6 shows the observed rainfall and inflow of the CR within the study area.

Monitoring nodes were selected to verify the status of the urban drainage network. The nodes where the first and maximum flooding occurred were selected as the monitoring nodes. Table 2 shows the results of the maximum flooding nodes when historical rainfall data from 2010 to 2019 were applied.

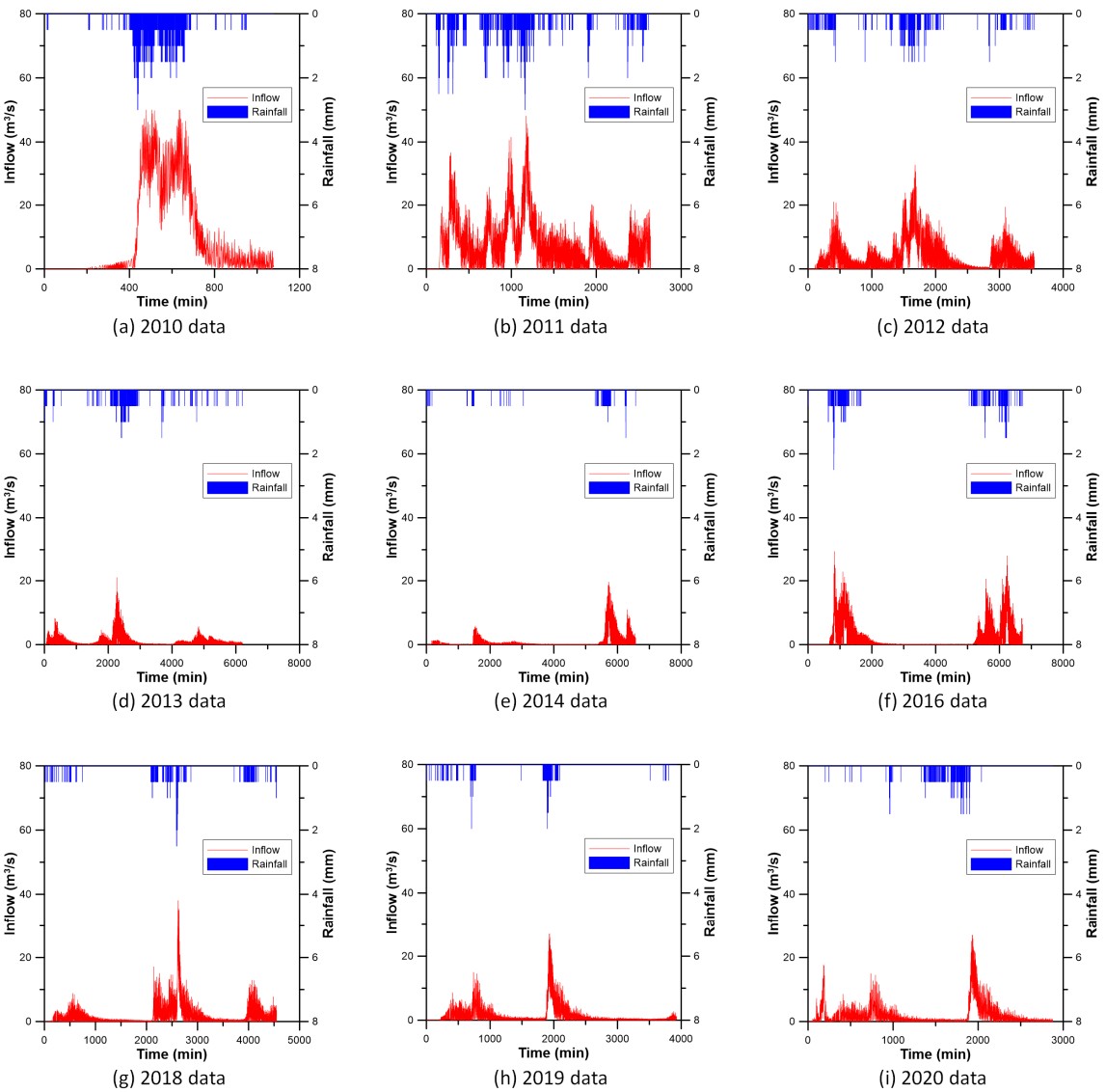

**Figure 6.** Observed rainfall and inflow of CR within the study area.

**Table 2.** Results of maximum flooding nodes.

| Rainfall Events | 2010 | 2011 | 2012 | 2013 | 2014 | 2016 | 2018 | 2019 |
|---|---|---|---|---|---|---|---|---|
| Maximum flooding nodes | 550 | 550 | 550 | 550 | 550 | 550 | 550 | 550 |

For all rainfall events, the maximum number of flooding nodes was 550. Node 550 was selected as the monitoring point. For the first flooding nodes, durations of 30, 60, and 90 min were applied, and synthetic rainfall data based on the Huff distribution were used. Table 3 lists the results for the first flooding nodes.

**Table 3.** Results of first flooding nodes.

| Duration (m) | 30 | 60 | 90 |
|---|---|---|---|
| First flooding nodes | 560 | 560 | 575 |

As the results listed in Table 3 indicate, as the first flooding nodes, node 560 was selected at 30 and 60 min, whereas node 575 was selected at 90 min. Nodes 560 and 575

were selected as the monitoring nodes. The water levels of three nodes (550, 560, and 575) were added to the input data. Data on the rainfall, water level of the monitoring nodes, and CR inflow were constructed as the input data.

### 3.3. Inflow Prediction Using MLPIHS

Before preparing for the training using the input data, a data preprocessing was conducted. A correlation analysis was applied to the input data for the training using MLPs, the results of which are listed in Table 4.

**Table 4.** Results of correlation analysis.

| Data Type | Monitoring Node (550) | Monitoring Node (560) | Monitoring Node (575) | Rainfall Data |
|---|---|---|---|---|
| Lag time (min) | 15 | 14 | 13 | 17 |
| Correlation coefficient | 0.813 | 0.949 | 0.952 | 0.747 |

As shown in Table 4, the lag time ranged from 13 to 17 min, which implies that the inflow is predictable after 13 min. Additionally, normalization was conducted to convert all input data for adjusting the data scale. The results of the MLP were obtained by applying Adadelta, Adagrad, Adam, Adamax, Ftrl, Nadam, RMSprop, and SGD provided by TensorFlow as optimizers. The performances of MLPHS and MLPIHS were compared. To calculate the error of each result, the mean square error (MSE) based on the square value was applied along with the mean absolute error (MAE) based on the absolute values. The MSE is expressed as shown in Equation (5):

$$MSE = \frac{\sum_{i=1}^{n}(x_o - x_i)^2}{n} \tag{5}$$

where $x_o$ is the observed data, $x_i$ is the simulated data, and $n$ is the number of data points. The MAE is expressed as shown in Equation (6):

$$MAE = \frac{\sum_{i=1}^{n}|x_o - x_i|}{n} \tag{6}$$

where $x_o$ is the observed data, $x_i$ is the simulated data, and $n$ is the number of data points. Table 5 shows the results of the inflow prediction.

**Table 5.** Results of inflow prediction.

| Method | Adadelta | Adagrad | Adam | Adamax | Ftrl | Nadam | RMSprop | SGD |
|---|---|---|---|---|---|---|---|---|
| MSE | 4.312000 | 5.970060 | 3.082933 | 3.106901 | 11.978707 | 3.199255 | 3.261781 | 6.689764 |
| MAE | 1.285771 | 1.261158 | 1.021378 | 1.024476 | 1.635111 | 1.036619 | 1.085643 | 1.443204 |
| Method | Adadelta +HS | Adagrad +HS | Adam +HS | Adamax +HS | Ftrl +HS | Nadam +HS | RMSprop +HS | SGD +HS |
| MSE | 3.199255 | 3.088522 | 3.071538 | 3.078417 | 2.936929 | 3.137013 | 3.034306 | 4.901860 |
| MAE | 1.036619 | 1.028809 | 1.018922 | 1.025179 | 1.001793 | 1.044988 | 1.015344 | 1.304466 |
| Method | Adadelta +IHS | Adagrad +IHS | Adam +IHS | Adamax +IHS | Ftrl +IHS | Nadam +IHS | RMSprop +IHS | SGD +IHS |
| MSE | 3.029370 | 2.984212 | 3.047000 | 3.046682 | 2.930192 | 3.119831 | 3.020394 | 3.247332 |
| MAE | 1.017907 | 1.020483 | 1.030821 | 1.024705 | 0.989532 | 1.037507 | 1.000169 | 1.177632 |

Among the MLPs to which the existing optimizers in Table 4 were applied, Adam showed the smallest error in terms of the MSE and MAE. The MSE of all optimizers except for Ftrl, which showed the largest error, did not exceed 10. For MLPHS, Ftrl showed the

smallest errors in terms of the MSE and MAE. Ftrl showed a large error for an MLP, whereas the MSE and MAE of Ftrl+HS decreased the most in MLPHS. For MLPIHS, Ftrl+IHS showed the smallest error in terms of the MSE and MAE, and the results were similar to those of Ftrl+HS. Ftrl+IHS showed a slightly improved accuracy compared with Ftrl+HS. Figure 7 shows a comparison of the results of Adam, Ftrl+HS, and Ftrl+IHS.

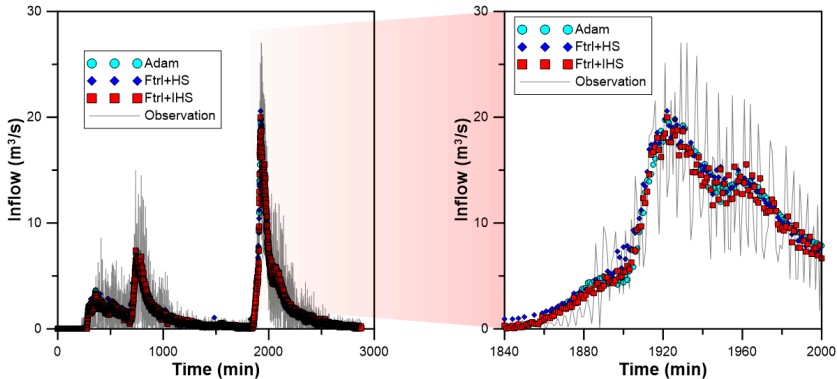

**Figure 7.** Comparison of prediction for each method.

As shown in Figure 7, the error for Adam was relatively large compared with those of Ftrl+HS and Ftrl+IHS. The difference at the peak value is similar for all three optimizers; however, after the peak value, Ftrl+IHS has a small difference from the observed value.

## 4. Discussion

The $R^2$ values for the observed and predicted inflows for each MLP were calculated. The value of $R^2$ is expressed as shown in Equation (7):

$$R^2 = \frac{\sum_{i=1}^{n}(x_i - \overline{x})^2}{\sum_{i=1}^{n}(x_o - \overline{x})^2} \tag{7}$$

where $x_o$ is the observed data, $x_i$ is the simulated data, $n$ is the number of data points, and $\overline{x}$ is the average of $x_o$. Additionally, a comparison was conducted based on the flow rates of the observed and predicted inflows. Figure 8 shows a comparison of $R^2$ for the observed and predicted inflows when using Adam.

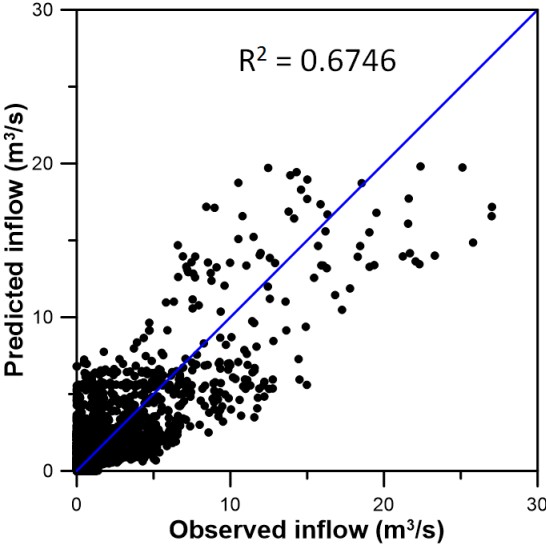

**Figure 8.** Comparison of $R^2$ for observed and predicted inflows when using Adam.

As shown in Figure 8, the MLP using Adam predicted a slightly higher value at a low inflow and a slightly lower value at a high inflow. Figure 9 shows a comparison of $R^2$ for the observed and predicted inflows when using Ftrl+HS.

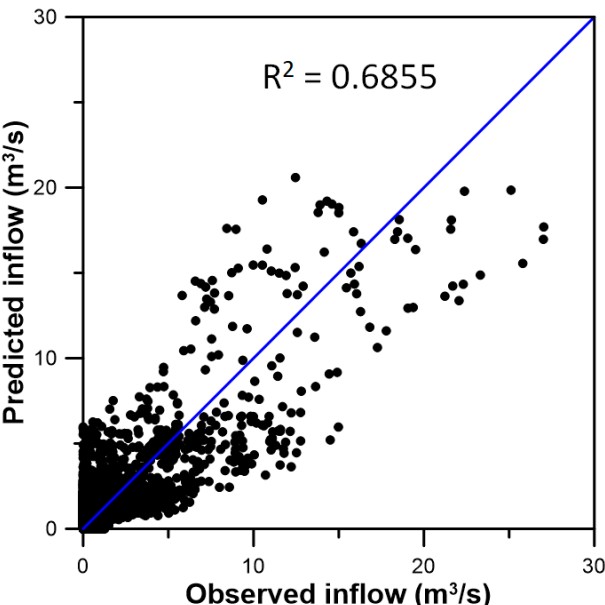

**Figure 9.** Comparison of $R^2$ for observed and predicted inflows when using Ftrl+HS.

As shown in Figure 9, the MLP using Ftrl+HS predicted a slightly higher value at a low inflow and a slightly lower value at a high inflow. With a slight difference, a similar pattern to the results in Figure 8 can be seen. Figure 10 shows a comparison of $R^2$ for the observed and predicted inflows using Ftrl+IHS.

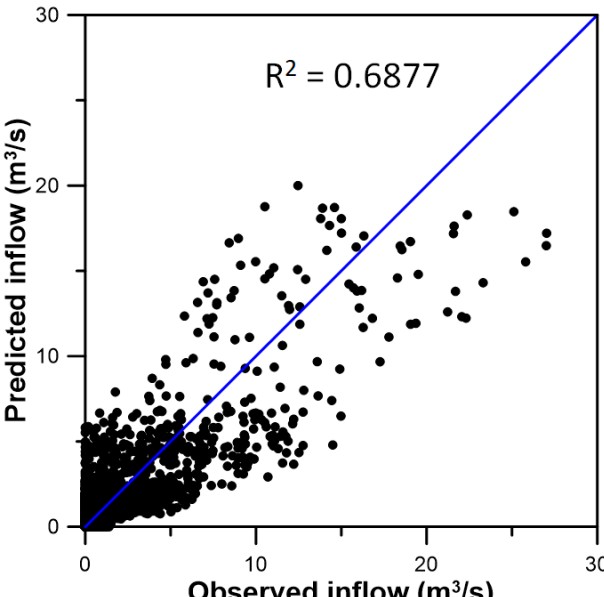

**Figure 10.** Comparison of $R^2$ for observed and predicted inflows when using Ftrl+IHS.

As shown in Figure 10, the MLP using Ftrl+IHS predicted a slightly higher value at a low inflow and a slightly lower value at a high inflow. With a slight difference, a similar pattern to the results shown in Figures 8 and 9 can be seen. Based on the results from Figures 8–10, all optimizers tend to predict slightly higher values at a low inflow and lower

values at a high inflow. However, the optimizers combining HS and IHS were improved compared with the existing optimizer.

In this study, a structure of MLP that can produce relatively good results was selected through results according to the number of nodes and the number of hidden layers was selected. Figure 11 shows the results according to the number of nodes (Adadelta and Adagrad).

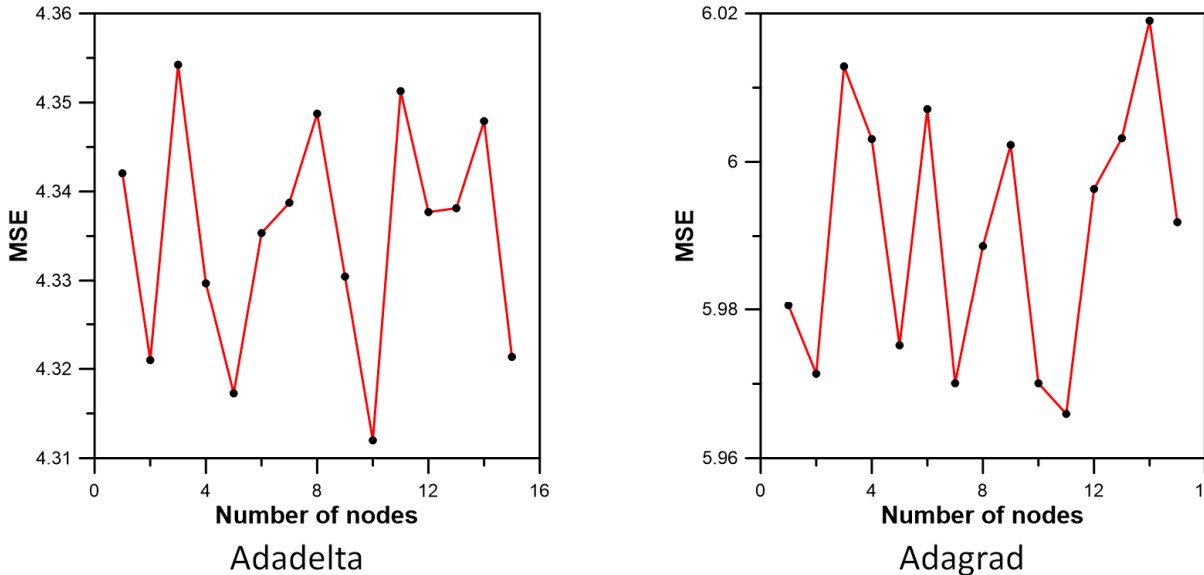

**Figure 11.** Results according to the number of nodes (Adadelta and Adagrad).

When Adadelta was applied in MLP, the best results were obtained there were 10 nodes. When Adagrad was applied in MLP, the best results were obtained there were 11 nodes. Figure 12 shows the results according to the number of nodes (Adam and Adamax).

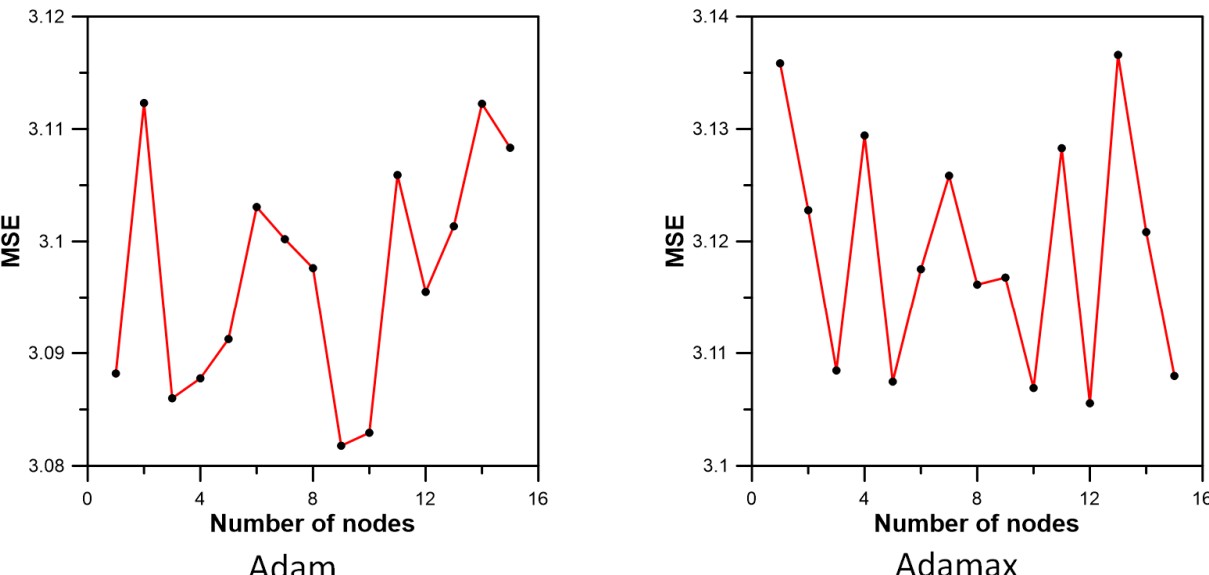

**Figure 12.** Results according to the number of nodes (Adam and Adamax).

When Adam was applied in MLP, the best results were obtained there were 9 nodes. When Adamax was applied in MLP, the best results were obtained there were 12 nodes. Figure 13 shows the results according to the number of nodes (Ftrl and Nadam).

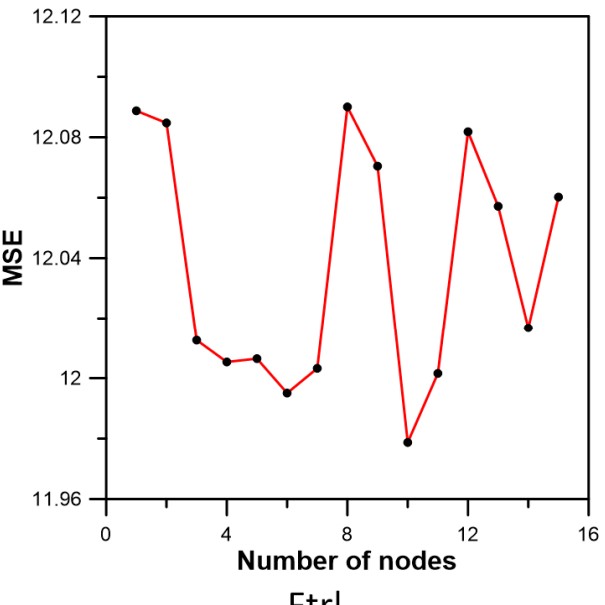
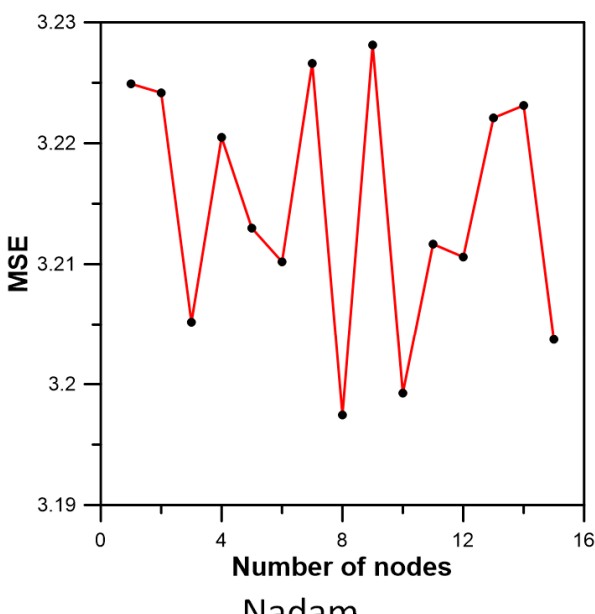

**Figure 13.** Results according to the number of nodes (Ftrl and Nadam).

When Ftrl was applied in MLP, the best results were obtained there were 10 nodes. When Nadam was applied in MLP, the best results were obtained there were 8 nodes. Figure 14 shows the results according to the number of nodes (RMSprop and SGD).

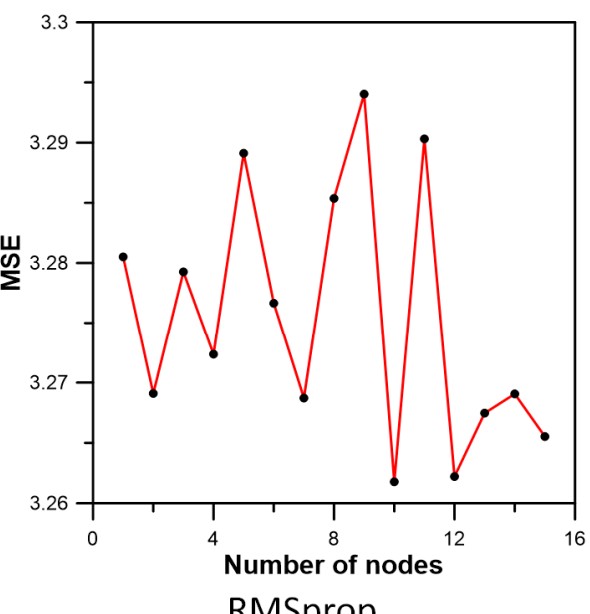
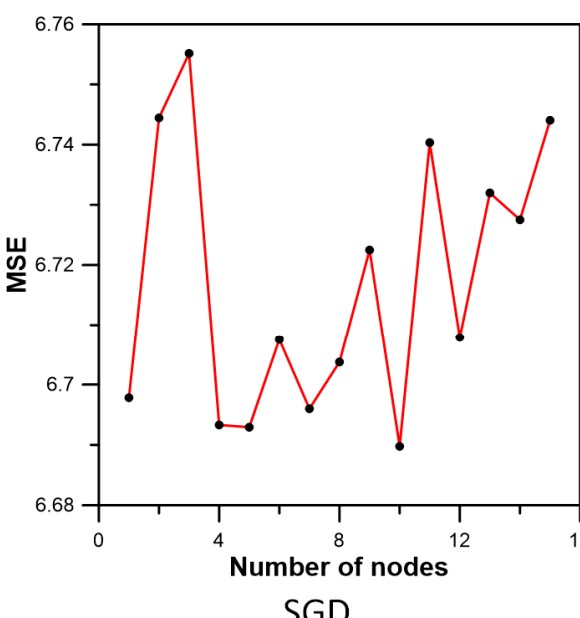

**Figure 14.** Results according to the number of nodes (RMSprop and SGD).

When RMSprop was applied in MLP, the best results were obtained there were 10 nodes. When SGD was applied in MLP, the best results were obtained there were 10 nodes. Figure 14 shows the results according to the number of nodes (RMSprop and SGD). Although there were differences for each optimizer, many results showed the smallest MSE when 10 nodes were applied. Figure 15 shows the results according to the number of hidden layers (Adadelta and Adagrad).

When Adadelta was applied in MLP, the best results were obtained there were 7 hidden layers. When Adagrad was applied in MLP, the best results were obtained there were

4 hidden layers. Figure 16 shows the results according to the number of hidden layers (Adam and Adamax).

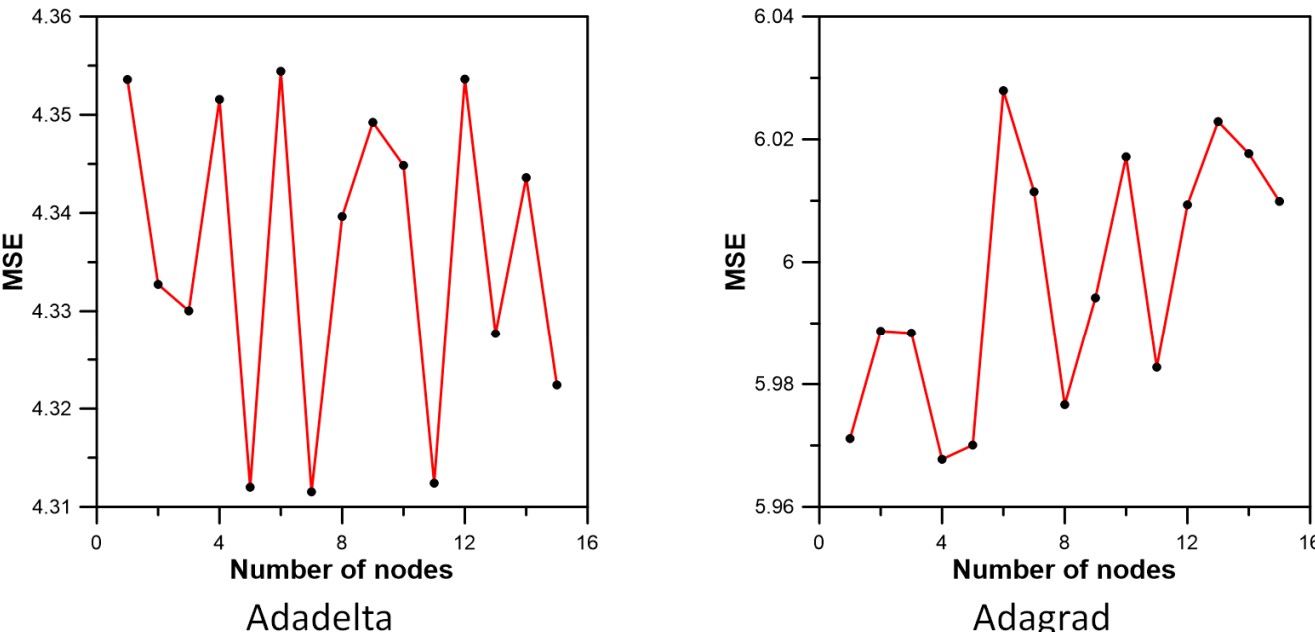

**Figure 15.** Results according to the number of hidden layers (Adadelta and Adagrad).

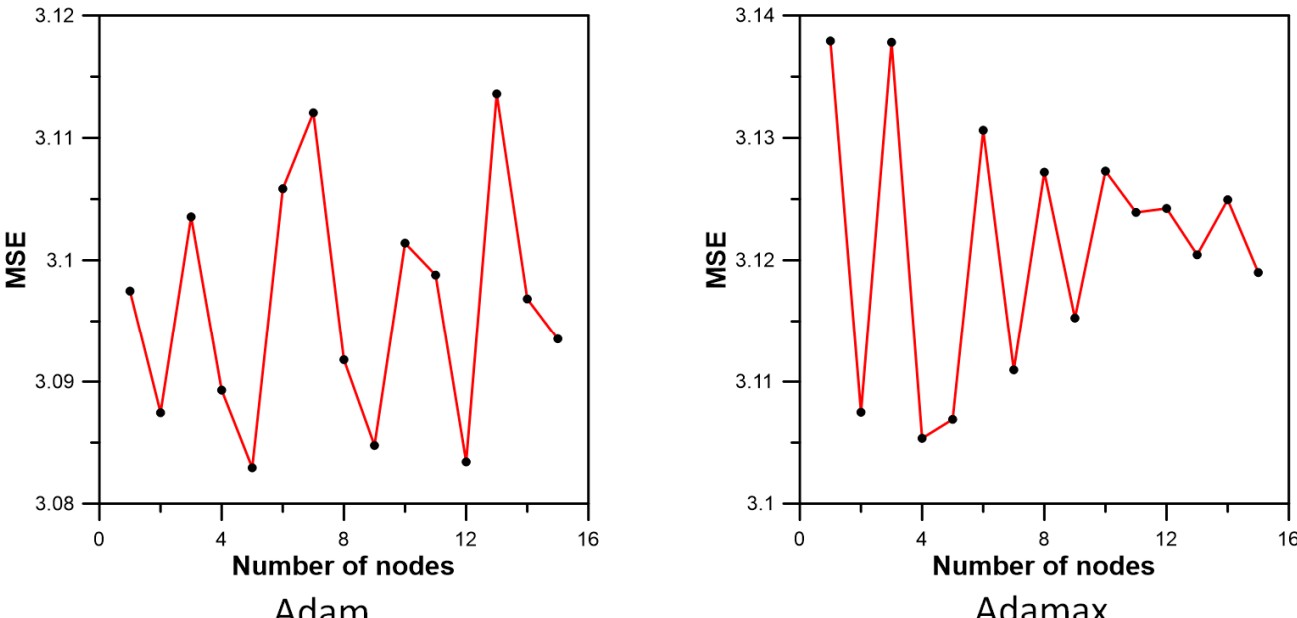

**Figure 16.** Results according to the number of hidden layers (Adam and Adamax).

When Adam was applied in MLP, the best results were obtained there were five hidden layers. When Adamax was applied in MLP, the best results were obtained there were four hidden layers. Figure 17 shows the results according to the number of hidden layers (Ftrl and Nadam).

When Ftrl was applied in MLP, the best results were obtained there were five hidden layers. When Nadam was applied in MLP, the best results were obtained there were seven hidden layers. Figure 18 shows the results according to the number of hidden layers (Ftrl and Nadam).

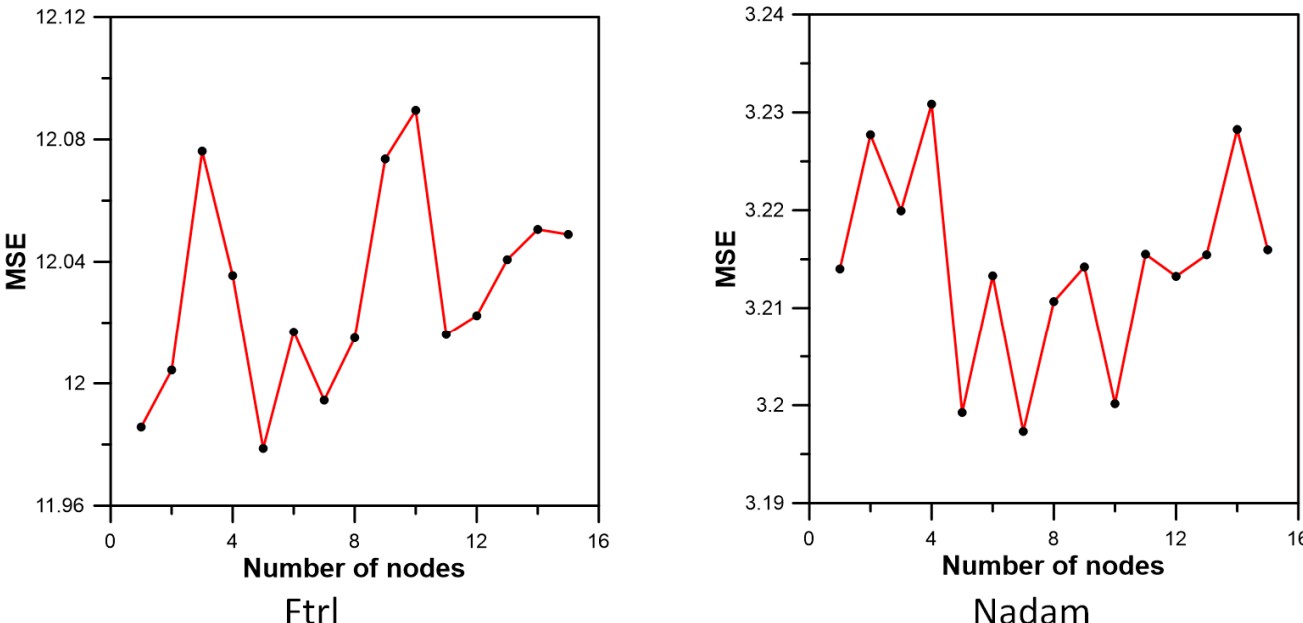

**Figure 17.** Results according to the number of hidden layers (Ftrl and Nadam).

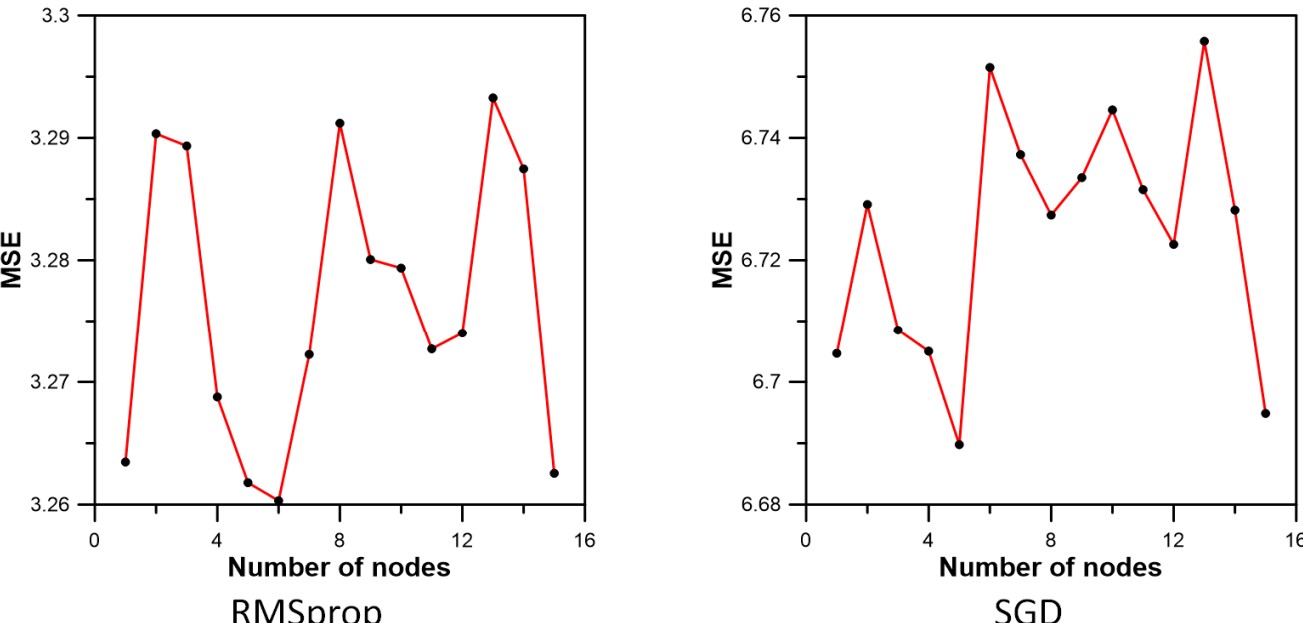

**Figure 18.** Results according to the number of hidden layers (RMSprop and SGD).

When RMSprop was applied in MLP, the best results were obtained there were five hidden layers. When SGD was applied in MLP, the best results were obtained there were seven hidden layers. Although there were differences for each optimizer, many results showed the smallest MSE when five hidden layers were applied. Based on the results according to the number of nodes and the number of hidden layers, 10 nodes and five hidden layers were selected.

In the case of a simpler structure of MLP, it could be difficult to show a small MSE value. Figure 19 shows the prediction results according to epochs when Ftrl+IHS is applied.

It is expected that the error could be reduced if training was conducted for more epochs. However, many epochs do not guarantee good results, as overfitting problem could occur.

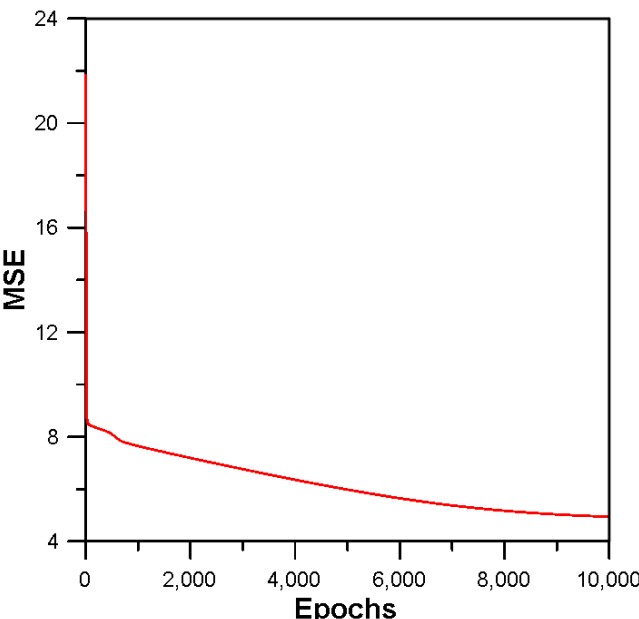

**Figure 19.** Prediction results according to the epochs when Ftrl+IHS is applied.

## 5. Conclusions

The important aspect of this study is the CR inflow prediction for the preemptive operation of an urban drainage facility such as a pump station. There are two disadvantages to the use of existing MLP optimizers. Existing MLP optimizers can fall into a local optimal solution because they are highly dependent on the initial values (weights and biases). Additionally, because existing optimizers lack storage space, the best result might not be stored during the learning process.

To overcome these two disadvantages, the use of existing optimizers combined with metaheuristic optimization algorithms such as HS and IHS was suggested. Prior to the CR inflow prediction, data preprocessing was conducted by applying a correlation analysis and normalization. To select monitoring nodes as training data, the first flooding nodes and the maximum flooding nodes were obtained. The comparison results showed that the MLP to which the existing optimizers in combination with HS and IHS were applied showed relatively accurate results in terms of both the MSE and MAE. The results revealed that MLPHS and MLPIHS predicted the inflow more accurately than the MLP with existing optimizers.

The model used for inflow prediction in this study runs in a continuous sequence from the beginning. Because prediction was run after training, training and prediction did not proceed simultaneously. However, real-time prediction could be possible if the learning model and the prediction model are executed simultaneously when real-time data is input.

Various follow-up studies can be conducted to overcome this lack of usability. To increase the usability of MLPHS and MLPIHS, an optimized MLP structure can be developed. If metaheuristic optimization algorithms including evolutionary algorithms are applied to optimize the structure (numbers of nodes and hidden layers), it will be possible to improve the usability of an MLP. If the method described in this study is combined with an approach to rainfall prediction, it will be possible to secure additional operational times for urban drainage facilities.

**Funding:** This study was funded by Basic Science Research Program through the National Research Foundation of Korea (NRF) funded by the Ministry of Education (NRF-2019R1I1A3A01059929).

**Data Availability Statement:** Data available on request due to restrictions (public management policy). The data presented in this study are available on request from the corresponding author with the permission of the public institution. The data are not publicly available due to public management policy.

**Acknowledgments:** This research was supported by Basic Science Research Program through the National Research Foundation of Korea (NRF) funded by the Ministry of Education (NRF-2019R1I1A3A01059929).

**Conflicts of Interest:** The author declares no conflict of interest.

## Abbreviations

| | |
|---|---|
| CR | Centralized reservoir |
| DR | Decentralized reservoir |
| UDS | Urban drainage system |
| MLP | Multilayer perceptron |
| ANN | Artificial neural network |
| RNN | Recurrent neural network |
| CNN | Convolutional neural network |
| LSTM | Long short-term memory |
| GRU | Gated recurrent unit |
| GA | Genetic algorithm |
| PSO | Particle swarm optimization |
| HS | Harmony search |
| IHS | Improved harmony search |
| RCGA | Real-coded genetic algorithm |
| MLPHS | MLP using new optimizer combined with HS |
| MLPIHS | MLP using new optimizer combined with IHS |
| GD | Gradient descent |
| SGD | Stochastic gradient descent |
| NAG | Nesterov accelerated gradient |
| Adagrad | Adaptive gradient |
| RMSprop | Root mean squared propagation |
| AdaDelta | Adaptive delta |
| Adam | Adaptive moment |
| Nadam | Nesterov accelerated adaptive moment |
| HMS | Harmony memory size |
| HMCR | Harmony memory considering rate |
| PAR | Pitch adjusting rate |
| BW | Bandwidth |
| HM | Harmony memory |
| SWMM | Storm water management model |
| MSE | Mean square error |
| MAE | Mean absolute error |

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
