# Peer review of "Inflow Prediction of Centralized Reservoir for the Operation of Pump Station in Urban Drainage Systems Using Improved Multilayer Perceptron Using Existing Optimizers Combined with Metaheuristic Optimization Algorithms"

_water, doi:10.3390/w15081543_

Round 1

Reviewer 1 Report

The introduction is a list of authors who have developed an optimizer or something similar. It is full of acronyms and does not provide a real guide to understanding the algorithms because each quotation fails to really present the algorithm to those who do not know it a priori. The methodology disregards the initial declaration of having experimented the application of the TensorFlow package (probably under Mathlab) with its 8 optimizers Agadelta, Agagrad... Probably paragraph 2.3.1 could contain a table of optimizers. For each of them there are provide very little explanation and thay may be in a table or better in a list. It is not clear why for such a simple case a neural network of 10 nodes with three hidden layers was sized. Perhaps a simpler network would have had easier training (<10000 epochs) and less risk of falling back into local optimal solutions. To size networks, scholars sometimes use evolutionary algorithms.

For an inattentive reader like me, the advantage, in terms of MSE and MAE, of Ftrl+HIS or Ftrl+HR vs Adam which is simpler is too small. R2 also does not change. For a simple old scholar like me, the conclusion is that in this application, it is completely useless  IHS.

Please check the abstract line 12 nonstructural.

If possible anticipate the reference to TensorFlow and its supporting literature in methods. At the beginning of methods if you agree.

In Tab 3 it is Rainfall data and not pump?

In line 274: what is historical rainfall for you?

In figure 5 check boundary of each subarea.

Author Response

I appreciate the reviewer’s helpful comments. Detailed descriptions of how these comments were addressed are provided below. Please note that additions/modification to the original paper are highlighted in red in the revised paper. Additional English editing was carried out and certificate of English editing was attached.

Reviewer 2 Report

The authors have prepared a paper that should be of interest to readers of Water.  However, some points to note are:

1 - Using the usual description of management strategies, a CR is a structural measure ad not a non-structural as described by the authors.

2 - There are random referencing styles throughout the paper.  Consistency in referencing is needed.

3 - In the discussion of normalisation of variables, the authors do not state how the max and min values were determined.  This is important as the ANN cannot predict flows outside the range of the normalisation - see Minns and Hall.

4 - There is a need for the authors to clarify if they are state updating the predictive model or running the predictive model as a continuous sequence from the start.  This has implications for the forecast flows and the reliability of forecast horizons.

Author Response

(The authors gave the same response as above.)
